# Protective Effects of Four Natural Antioxidants on Hydroxyl-Radical-Induced Lipid and Protein Oxidation in Yak Meat

**DOI:** 10.3390/foods11193062

**Published:** 2022-10-02

**Authors:** Sining Li, Shanhu Tang, Jinjin Li, Lamei Chen, Yuan Ma

**Affiliations:** College of Food Science and Technology, Southwest Minzu University, Chengdu 610041, China

**Keywords:** natural antioxidant, hydroxyl radical, protein oxidation, lipid oxidation, yak meat

## Abstract

The impacts of natural antioxidants, including ferulic acid, diallyl sulfide, α-tocopherol, and rutin, at a level of 0.2 g/kg on lipid and protein oxidation of minced yak meat in a hydroxyl-radical-generating system were investigated, and the effectiveness was compared with synthetic antioxidant 2,6-di-tert-butyl-4-methylphenol (BHT). The exposure of yak meat to oxidative stress from 12 h to 24 h elevated lipid and protein oxidation. Treatments with antioxidants resulted in significantly lower peroxides, conjugated dienes, and thiobarbituric acid-reactive substances, and were also effective in retarding the formation of carbonyl groups, reducing the loss of sulfhydryl groups and protecting α-helix contents, of which ferulic acid and rutin were the most effective. Myosin heavy chain underwent lower degradation in the samples treated with ferulic acid or rutin compared with the oxidized control and other antioxidant treatments, while that of the BHT treatment showed a similar intensity with oxidized control at 24 h of oxidation. The physical stability of myofibrillar proteins in samples with antioxidants from high to low was rutin, ferulic acid, α-tocopherol, and BHT~diallyl sulfide. These results indicate that rutin and ferulic acid may be promising antioxidants in inhibiting the oxidative reactions during the processing of yak meat.

## 1. Introduction

Animal muscles enriched with lipids and proteins are easily oxidized, and the oxidative products potentially affect the nutritional, physicochemical, and organoleptic properties of meat products [1]. However, lipid and protein oxidation are inevitable in meat processing, transportation, and storage. It is reported that lipid oxidation, being a major cause of off flavor and discoloration in meat and meat products, can result in a product that is undesirable for human consumption [2,3], and protein oxidation occurs simultaneously with lipid oxidation in meat systems, which leads to tenderness reduction, discoloration, flavor deterioration, and functionality change of the product [4]. Therefore, it is critical to control the extent of lipid and protein oxidation during meat product processing.

Numerous research works have suggested that the lipid and protein oxidation in processed meat products can be retarded or inhibited by the supplementation of commercial synthetic or natural antioxidants [5,6,7]. Synthetic antioxidants, such as butylated hydroxyanisole (BHA), 2,6-di-tert-butyl-4-methylphenol (BHT), and tert-butylhydroquinone (TBHQ), have been associated with potential safety problems due to their health risks and toxicity, which have restricted their applications in industrial processing. Consequently, the utilization of naturally occurring substances is gaining popularity as alternatives to the synthetic additives. For instance, tea polyphenols have been used extensively as antioxidants in the food industry. Their antioxidant effect has also been confirmed to be comparable to that of synthetic antioxidants. Many studies have confirmed that some natural compounds were effective in inhibiting lipid oxidation of muscle foods [5,8]. For a large number of natural compounds, such as α-tocopherol and some polyphenols, their roles in retarding protein oxidation during the preparation of meat and meat products have also been demonstrated [6,9]. Ferulic acid, known as coniferic acid, is one of the phenolic acids, commonly found in fruits and vegetables, and its effectiveness against free radicals has been studied extensively [10,11]. Rutin, a flavonoid glycoside compound, is presented in asparagus and citrus fruits, and its antioxidant effectiveness is also evaluated [7]. It was reported that polyphenols have double-sided roles of anti-oxidation and pro-oxidation in foods, which mainly depend on the type and dosage of the polyphenols used [12,13]. Diallyl sulfide is an organosulfur compound derived from *Allium* plants, such as garlic and onion, which is a potential compound for enhancing lipid stability [14]. However, uses of diallyl sulfide for inhibiting protein oxidation in muscle food systems have not been reported to date.

Reactive oxygen species (ROS), such as hydroxyl radicals (HO•), peroxy radicals (ROO•), and superoxide anions (O_2_^−^•), are the main substances that induce oxidation of lipids and proteins. Since hydroxyl radical is one of the most active free radicals, it has been used to study the mechanisms of lipid and protein oxidation involved in complex muscle food systems [13,15,16]. Usually, the hydroxyl radical is produced via the Fenton reaction in the presence of hydrogen peroxide, reduced transition metal, and reductants [17]. Yak meat contains relatively higher concentrations of myoglobin [18] and polyunsaturated fatty acid [19] than yellow cattle meat, which makes it more susceptible to oxidation. It is important to evaluate the applicability of different natural antioxidants for alleviating oxidation in yak meat.

However, only a few studies have been conducted exploring the potentials of natural antioxidants in preventing the oxidation of lipids and proteins and extending the shelf-life of yak meat products. The effect and the mechanism of hydroxyl radicals on their antioxidant properties of natural antioxidants against lipid and protein oxidation of yak meat have not been elucidated. Our previous studies showed that pomegranate peel extract and garlic powder had protective effects against lipid and protein oxidation of yak meat [20,21]. Garlic powder and pomegranate peel extract are rich in diallyl sulfide and ferulic acid, respectively, and the objectives of our present work were to investigate the potential effectiveness of four natural antioxidants, including ferulic acid, diallyl sulfide, α-tocopherol, and rutin, in protecting yak meat lipids and proteins from oxidation in a constructed hydroxyl radical oxidation system and to evaluate the antioxidant effectiveness in comparison with the synthetic antioxidant BHT. Our findings are expected to guide the development of new strategies for preventing the lipids and proteins in yak meat from oxidation.

## 2. Materials and Methods

### 2.1. Chemicals

Chemicals, including BHT (purity 98%), ferulic acid (purity 97%), diallyl sulfide (purity 98%), α-tocopherol (purity 96%), and rutin (purity 98%), were purchased from Macklin Biotechnology Co., Ltd. (Shanghai, China). 2-Thiobarbituric acid (TBA), 1,1,3,3-tetraethoxypropane, 2,4-dinitrophenylhydrazine (DNPH), 5,5′-dithio bis (2-nitrobenzoic acid) (DTNB), and bovine serum albumin were obtained from Sigma-Aldrich (St. Louis, MI, USA). Other chemicals and reagents were of analytical grade and were provided by Kelong Company (Chengdu, China).

### 2.2. Sample Preparation

Fresh yak *Longissimus thoracis* muscles were obtained after 24 h postmortem from a commercial abattoir (New Hope Yak Industry Co. Ltd., Hongyuan, Sichuan Province, China). The connective tissue was removed; then, the muscles were minced using a meat grinder with a 5 mm plate (Midea, China). Muscles without or with antioxidant BHT (positive control), ferulic acid, diallyl sulfide, α-tocopherol, or rutin were subjected to the oxidation in a constructed hydroxyl radical system containing 10 µmol/L FeCl_3_, 0.1 mmol/L ascorbic acid, and 10 mmol/L H_2_O_2_ in 20 mmol/L phosphate buffer (pH 7.0). The individual antioxidant was added at a proportion of 0.2 g/kg and mixed well into its corresponding minced portion using a spatula for 1 min to ensure even distribution. Subsequently, each minced muscle was oxidized in a hydroxyl-radical-generating phosphate buffer, and the ratio of muscle sample to buffer was 1:2 (*w*/*v*). The sample without oxidants (FeCl_3_, ascorbic acid, and H_2_O_2_) was used as the non-oxidized control. All treatments were incubated at 4 °C for 12 h or 24 h. The oxidation was terminated with 1 mmol/L EDTA (final concentration). The samples were centrifuged at 8000× *g* for 15 min at 4 °C to remove the hydroxyl radicals.

### 2.3. Lipid Peroxide (POV) Determination

The POV was evaluated according to the method of Huang et al. [22]. Briefly, 2 g of samples was homogenized at 12,000 rpm for 30 s with 15 mL of chloroform: methanol (2:1, *v*/*v*). Then, 3 mL of 0.5% (*w*/*v*) NaCl was added, and the mixture was vortexed for 15 s and centrifuged at 3000× *g* for 10 min at 4 °C. The lower phase (5 mL) was collected and mixed well with 5 mL of chloroform: methanol (2:1, *v*/*v*); then, 25 μL of 30% (*w*/*v*) ammonium thiocyanate and 25 μL FeCl_2_ (0.4 g barium chloride was dissolved in 50 mL of deionized water, followed by addition of an equal volume of ferrous sulfate solution prepared by dissolving 0.5 g ferrous sulfate in 50 mL of deionized water) were added to the assay for POV. The mixture was incubated for 5 min at room temperature (20 ± 2 °C), and the absorbance was measured at 500 nm using a spectrophotometer (Aoyi, Shanghai, China). A standard curve was prepared using reduced iron. A blank contained all the reagents, except the sample. The POV was expressed as meq/kg sample.

### 2.4. Conjugated Diene (CD) Determination

The CD measurement adopted the method described by Juntachote et al. [23]. In brief, 2 g of the sample was suspended in 20 mL distilled water and homogenized at 12,000 rpm for 30 s to form a smooth slurry. A 2 mL aliquot of this suspension was mixed with 20 mL extracting solution (hexane: isopropanol = 3:1, *v*/*v*) for 1 min and centrifuged at 3000× *g* for 5 min at 4 °C. The absorbance of the supernatant was read at 233 nm by a spectrophotometer (Aoyi, Shanghai, China). The CD was calculated by the molar extinction coefficient of 25,200 L/(mol•cm) and expressed as mmol/kg sample.

### 2.5. Thiobarbituric Acid–Reactive Substance (TBARs) Assay

The levels of TBARs were determined according to the method of Berardo et al. [24]. Each portion of the sample (5 g) with 15 mL of 7.5% (*w*/*v*) TCA solution containing 0.1% (*w*/*v*) gallic acid and 0.1% (*w*/*v*) EDTA was homogenized at 8500 rpm for 30 s, and then, the mixture was incubated at 70 °C for 30 min. Subsequently, the homogenate was filtered through a filter paper, and the filtrate (3.5 mL) with 20 mmol/L TBA solution (2.5 mL) was incubated in a boiling water bath for 40 min. After cooling, the absorbance was measured at 532 nm using a spectrophotometer (Aoyi, Shanghai, China). TBARs values were calculated from a standard curve of 1,1,3,3-tetraethoxypropane [25] and expressed as mg malondialdehyde (MDA)/kg sample.

### 2.6. Myofibrillar Protein (MP) Extraction

The MP extracts were obtained following the method of Zhang et al. [26], with minor modifications. The samples were homogenized at 10,000 rpm for 60 s with 20 mmol/L phosphate buffer (1:4, *w*/*v*) containing 0.1 mol/L NaCl, 2 mmol/L MgCl_2_, and 1 mmol/L EGTA (pH 7.0). Afterward, the homogenate was centrifuged at 2500× *g* for 15 min (4 °C), and the supernatant was discarded. The pellet was washed twice with the buffer using the same homogenate and centrifugation conditions as indicated above. The pellet was then washed three times with 0.1 mol/L NaCl at a 1:4 ratio (*w*/*v*) in sequence. After discarding the supernatant, the resulting pellet was dissolved in 8 volumes of 20 mmol/L phosphate buffer (pH 6.0), filtered through four layers of cheese cloth, and centrifuged at 2500× *g* (4 °C) to obtain the MPs.

### 2.7. Carbonyl Group Determination

The carbonyl group was determined according to a previously reported method, with slight alterations [27]. MPs were diluted to 5 mg/mL with 20 mmol/L phosphate buffer (0.6 mol/L NaCl, pH 6.0). One aliquot of 1 mL of MPs was reacted with 1 mL of 10 mmol/L DNPH in 2 mol/L HCl or 2 mol/L HCl (control) in the dark for 40 min at room temperature (20 ± 2 °C). Afterward, the mixture was precipitated with 20% TCA (*w*/*v*) and centrifuged at 12,000× *g* for 5 min (4 °C). The pellet was washed three times with 1 mL of ethanol/ethyl acetate solution (1:1, *v*/*v*) to eliminate DNPH, and the mixture was centrifuged at 8000× *g* for 5 min (4 °C). The precipitate was then dissolved in 3 mL of 6 mol/L guanidine hydrochloride and incubated at 37 °C for 15 min. After centrifugation at 12,000× *g* for 15 min (4 °C), the absorbance of the supernatant was measured at 370 nm, and the carbonyl content was calculated by the molar extinction coefficient of 22,000 L/(mol•cm).

### 2.8. Sulfhydryl Group Determination

The sulfhydryl group was determined by the method of Ellman [28]. A portion of 1 mL of MPs (5 mg/mL) was added to 8 mL Tris buffer (8 mol/L urea, 0.086 mol/L Tris, 0.09 mol/L glycine, 4 mmol/L EDTA, pH 8.0) and centrifuged at 8000× *g* for 15 min (4 °C). Then, 4.5 mL of the supernatant was mixed with 0.5 mL of 10 mmol/L DTNB and incubated at room temperature (20 ± 2 °C) for 30 min. A blank control was prepared by replacing the homogenate with Tris buffer (pH 8.0). The absorbance was measured at 412 nm, and the sulfhydryl content was calculated by the molar extinction coefficient of 13,600 L/(mol•cm).

### 2.9. Fourier Transform Infrared (FTIR) Spectroscopy

The spectroscopy was determined using the method described by Gangidi et al. [29], with some modifications. Each spectrum from a wave number ranging from 650 to 4000 cm^−1^ was acquired by FTIR spectroscopy (Perkin Elmer, Waltham, MA, USA) equipped with ATR prism crystal accessory, with an average of 32 scans at a spectral resolution of 4 cm^−1^. Measurements were performed using 1 mg/mL MP solution, which was placed on the surface of the ATR crystal, followed by pressure with a flat-tip plunger. Spectra in the range of 1600–1700 cm^−1^ were used to analyze the secondary structure of the samples [30].

### 2.10. SDS-PAGE Analysis

The SDS-PAGE analysis was performed with a 12% acrylamide resolving gel and a 5% acrylamide stacking gel [31]. The MP solution (4 mg/mL) was mixed at a ratio 4:1 with 250 mmol/L Tris-HCl buffer (pH 6.8) containing 10% (*w*/*v*) SDS, 5% (*v*/*v*) β-mercaptoethanol, 50% (*v*/*v*) glycerol, and 0.5% (*w*/*v*) bromophenol blue, and denatured at 95 °C for 5 min. After centrifugation at 10,000× *g* for 10 min, 10 μL of the supernatant was loaded onto each well, and the separation was conducted at 80 V for 30 min and then at 120 V for 90 min using a Mini-Protein electrophoresis system (Bio-Rad, CA, USA). After electrophoresis, the gel was stained in 0.1% (*w*/*v*) Coomassie Blue R-250 and subsequently de-stained overnight in a solution of 7.5% (*v*/*v*) methanol and 7.5% (*v*/*v*) acetic acid.

### 2.11. Physical Stability Measurement

The physical stability of the MPs was measured with the LUMiSizer^®^ (L.U.M. GmbH, Berlin, Germany), according to the method published by Tian et al. [32], with some modifications. The MP dispersions were slowly added to the bottom of the standard cuvette (2 mm disposable polyamide sample cuvette with a rectangular cross-section). The cuvette was placed horizontally into the instrument after capping and exposed to centrifugal force under 3500 rpm at 25 °C. The sample profiles were recorded at intervals of 20 s. The instability index, clarification as function of time divided by the maximum clarification possible, was calculated using the SEPView 6 software.

### 2.12. Statistical Analysis

All experiments were performed in triplicate, and the results were expressed as the mean ± standard deviation (SD). A *p*-value of less than 0.05 was selected as significant. Data were analyzed with one-way analysis of variance, and the possible differences between the antioxidant treatments were performed with Duncan multiple range tests. The significant differences between the values of the two time points were analyzed with a t-test. SPSS 21.0 software (IBM, Chicago, IL, USA) was used for the data analysis.

## 3. Results and Discussion

### 3.1. POV and CD Value

The POV represents the hydroperoxides (primary oxidation products) produced by the lipid oxidation process, and another primary compound of lipid oxidation is CD, formed after bisallylic hydrogen is uprooted by the reordering of the double bond, which leads to the generation of a conjugated double bond [33,34].

It is obvious to see that the POV values of minced yak meat significantly increased with the extension of the oxidation time (*p* < 0.05, Table 1), as evidenced by the higher POV at 24 h. The POV value of yak meat that was exposed to a hydroxyl-radical-generating system increased (*p* < 0.05), that is, from 0.11 to 0.24 meq/kg at 12 h, and 0.18 to 0.34 meq/kg at 24 h, respectively. As expected, the addition of an antioxidant could prevent the formation of hydroperoxides, which suggested that the initiation and propagation of lipid oxidation were more pronounced in the oxidized control when compared with the antioxidant-treated samples. Similar findings were reported by Maqsood et al. [5] who noted that the control samples contained higher POV than samples supplemented with natural antioxidants. There was no significance (*p* > 0.05) between natural antioxidants and BTH in preventing POV formation of minced yak meat incubated at 12 h in a Fe^2+^/H_2_O_2_ mixture, while ferulic acid and rutin demonstrated to be more active antioxidants (*p* < 0.05) than BHT, α-tocopherol, and diallyl sulfide in inhibiting the production of hydroperoxides in yak meat at 24 h.

Meanwhile, yak meat subjected to hydroxyl radical oxidation also resulted in increased CD concentration (*p* < 0.05), and the values in non-oxidized and oxidized controls were 0.47–0.51 mmol/kg and 0.78–0.88 mmol/kg, respectively. The double bonds of hydroperoxides in polyunsaturated fatty acids tend to be rearranged, which resulted in CD formation [35]. This indicated that hydroxyl radicals accelerated the rearrangement of double bonds in the hydroperoxides by the considerable impact of free radicals on oxidation reactions. Except for the non-oxidized control, oxidized control, and α-tocopherol treatment, the oxidation time also promoted the production of CD (*p* < 0.05). The positive effects (*p* < 0.05) of the use of antioxidants regarding the prevention of CD formation were obtained for which the values of CD were lower in ferulic acid and rutin treatments. Treatment with α-tocopherol had a similar effect as with BHT and diallyl sulfide in delaying the production of the CD value (*p* > 0.05). According to Öztürk-Kerimoğlu et al. [34], the CD content has a less determinative effect compared to other oxidation parameters. Thus, we should evaluate the lipid oxidation in combination with other indicators.

### 3.2. TBARs Content

Figure 1 illustrates the changes in the secondary lipid oxidation compounds (TBARs) in minced yak meat samples with and without the added antioxidants during oxidation. TBARs values of all samples increased with the extension of oxidation time from 12 to 24 h (*p* < 0.05), and a higher formation of TBARs in oxidized control was observed compared with non-oxidized control (*p* < 0.05). This event is due to hydroxyl radical attacks on the unsaturated fatty acid chain, giving rise to the formation of various 2-TBA reactive substances [10]. The results also indicated that the formation of TBARs was significantly inhibited with the presence of antioxidants during oxidation (*p* < 0.05). In contrast with synthesized antioxidant BHT, ferulic acid and rutin had stronger resistant effects on lipid oxidation (*p* < 0.05), while diallyl sulfide and α-tocopherol showed to be less effective (*p* < 0.05). It was possible to rank the natural antioxidant effectiveness on lipids as ferulic acid > rutin > α-tocopherol > diallyl sulfide, since this was the order of efficacy in preventing the formation of secondary lipid oxidation compounds. Wang et al. [6] also observed that polyphenols could retard lipid oxidation more effectively than α-tocopherol. The effects of phenolic compounds can be ascribed to their ability to scavenge •OH and additionally sequester prooxidative metal ions [36,37]. Interestingly, regardless of 12 h or 24 h of incubation, similar values (*p* > 0.05) between non-oxidized control and ferulic acid treatment were observed, implying lipid oxidation was almost completely inhibited by ferulic acid. This can be attributed to the fact that polyphenols can independently inhibit lipid oxidation through a free radical scavenging mechanism [6,38]. Notably, the highest TBARs values occurred in diallyl sulfide samples among the antioxidant treatments. On the contrary, Yin et al. [14] reported that diallyl sulfide showed significantly greater resistance to lipid oxidation than α-tocopherol. Nevertheless, the aforementioned study was conducted in ground beef during storage at 15 °C; the diallyl sulfide dosage used in the meat treatment was extremely low; and the preparation procedure of beef samples was not described in detail. Perhaps the different method employed for sample treatment led to this inconsistency in the events that occurred in the oxidation system constructed in vitro.

### 3.3. Carbonyl Content

Amino acids that contain NH or NH_2_ groups are extremely susceptible to being modified by oxidants into carbonyl derivatives; hence, protein carbonyl groups have been widely employed to assess the extent of oxidative modification of the proteins [37,39]. As shown in Figure 2, the results display that the carbonyl contents of hydroxyl-radical-generating system treated samples rose significantly (*p* < 0.05) with the extension of incubation time. The carbonyl contents of non-oxidized controls incubated for 12 h and 24 h were 1.48 and 1.71 nmol/mg protein, respectively, while the carbonyl contents of samples exposed to •OH (oxidized control) experienced rapid elevation, up to 1.8-fold (2.72 nmol/mg protein) at 12 h and 2.2-fold (3.77 nmol/mg) at 24 h. This increase in carbonyl groups, however, was significantly alleviated (*p* < 0.05) by the supplementation of antioxidants, which could be ascribed to the abilities of these antioxidants to quench free radicals. Among the five antioxidants, rutin, serving as a phenolic compound, was the most effective in inhibiting protein carbonyl formation, followed by ferulic acid, which showed similar carbonyl values as rutin during the oxidation. As Cheng et al. [7] and Kanski et al. [10] reported, polyphenols markedly suppressed protein carbonylation. Nevertheless, this was in contradiction with the observation of Wang et al. [6] who demonstrated that α-tocopherol at a level of 1 g/kg of the sample showed a higher inhibitory effect for fish mince compared to polyphenols. This conflict might be attributed to the different levels of antioxidants in muscle food systems. In our study, BHT showed similar effects as diallyl sulfide and α-tocopherol did during the first 12 h but lost its antioxidative effect thereafter, and this indicated that BHT was not a long-term effective antioxidant against protein oxidation. This may be due to the depletion and oxidation of BHT during the time period [40]. In agreement with our findings, Pedraza-Chaverrí et al. [41] also revealed diallyl sulfide effectively inhibited protein carbonylation in rats in vivo. The effect of oxidative stability for diallyl sulfide could be associated with its ability to ameliorate the oxidative stress [41]. Our results affirmed that the four natural antioxidants showed good activity in retarding protein oxidation during the 24 h of oxidation.

### 3.4. Sulphydryl Group

Sulfhydryl groups are one of the most reactive functional groups in proteins, and sulfhydryl groups of sulfur-containing amino acids are readily oxidized by hydroxyl radicals, forming intra- or intermolecular disulfide bonds [7]. The reduction in sulfhydryl groups is related to the protein oxidation in muscle foods. As shown in Figure 3, the protein sulfhydryl groups in minced yak meat decreased (*p* < 0.05) for the duration of oxidation from 12 h to 24 h. The sulfhydryl content of non-oxidized control was 203.48 and 193.33 nmol/mg protein for 12 h and 24 h of incubation. Upon •OH oxidation, the sulfhydryl content in oxidized control decreased by 11.56% (12 h) and 20.54% (24 h), respectively. All the treatments with antioxidants significantly relieved (*p* < 0.05) the loss of sulfhydryl groups compared to the oxidized control, which is consistent with the results of Wang et al. [6]. The higher sulfhydryl groups in the samples supplemented with antioxidant could be attributed to the capability of the antioxidant to scavenge and/or compete with free radicals [42]. Specifically, rutin provided the most substantial protection against the sulfhydryl group oxidation. Cheng et al. [7] revealed that rutin at a level of 0.32 g/kg sample exhibited a stronger ability in reducing the loss of sulfhydryl groups compared with the control. Similarly, Viskupicova al. [43] also claimed that rutin prevented sulfhydryl group oxidation in sarcoplasmic reticulum, whereas Guo et al. [13] reported that the addition of rutin could not prevent the loss of sulfhydryl groups in myofibrillar protein of *Coregonus peled*. These differences were largely affected by the dose used, as rutin was considered to be a pro-oxidant at higher doses [12,13].

### 3.5. Secondary Structure

The proportions of α-helix, β-sheet, β-turn, and random coil in MPs calculated by analyzing the amide I spectra are as plotted in Figure 4. For the non-oxidized control, the main structures resulted from β-sheet, α-helix, and β-turn; these corresponded to the percentage areas of around 42.77%, 26.37%, 19.52% at 12 h of incubation, and 39.37%, 24.76%, 23.38% at 24 h, respectively. The secondary structure contents in oxidized control demonstrated that α-helix decreased from 19.67% (12 h) to 13.58% (24 h), and β-sheet declined from 38.39% to 30.71%, while β-turn increased slightly from 25.99% to 27.86%, and random coil was elevated from 15.95% to 27.85% with the extension of oxidation time from 12 to 24 h. These results confirmed a previous finding that protein oxidation reduced the percentage of α-helix and β-sheet conformation while raising the random coil content [13], as oxidation may disrupt intramolecular hydrogen bonds, and hence, leads to the cleavage of α-helix and β-sheet and the generation of β-turn and random coil [44]. Nevertheless, these are partly inconsistent with the results reported by Zhu et al. [16] who stated that the protein oxidation lessened α-helix content and raised β-sheet proportion in the oxidation system. Zhu et al. [16] speculated that the α-helix was gradually transformed into β-sheet during the oxidation process.

Minced yak meat incorporated with an antioxidant demonstrated higher values of α-helix than the oxidized control, regardless of the oxidation time. A similar phenomenon was also observed in another study [7]. These results illustrated that the added antioxidants enhanced the protein structure stability. During oxidation from 12 h to 24 h, ferulic acid and rutin treatments produced relatively high α-helix content among the antioxidant-treated samples, which was ascribed to the interaction between the polyphenols and proteins through covalent bonds, such as hydrogen bonds [7,45]. At 24 h of oxidation, diallyl sulfide displayed relatively low α-helix content (19.57%) but relatively high β-sheet content (38.21%) among the natural antioxidants. It was believed that the appropriate extent of protein unfolding and conversion of α-helix to β-sheet were beneficial for gelation [30]. Therefore, further studies are suggested to clarify the effects of diallyl sulfide on the myofibrillar proteins of yak meat.

### 3.6. SDS-PAGE

The MP electrophoretograms of minced yak meat after 12 or 24 h of incubation in the Fe^2+^/H_2_O_2_ mixture system are presented in Figure 5. The electrophoretic profile showed a dramatic decrease in the intensity of almost all bands in the oxidized control compared with the non-oxidized control. Previous studies have also reported the considerable decrease in the intensity of the protein bands due to hydroxyl radicals [15,17]. This was likely caused by the cross-linking and aggregation of protein fragments due to protein oxidation [17]. The intensity of protein bands in both the oxidized control and antioxidant-treated samples also weakened with the extension of incubation time.

As shown in Figure 5, compared to the oxidized control, the addition of antioxidants showed remarkable increase in band intensity of myosin heavy chain (MHC) and proteins around 100 kDa, regardless of the oxidation time. These changes were closely related to the antioxidants used. The most remarkable alteration in protein bands was noted in BHT samples, in which the MHC was severely weakened at 24 h of incubation. For the samples treated with natural antioxidants, MHC bands from diallyl sulfide and α-tocopherol treated samples underwent much more degradation compared to ferulic acid and rutin treated samples. Ferulic acid and rutin, probably due to their antioxidant activity, retarded MPs degradation of yak meat. The active degradation of proteins in the oxidized control and those treated with BHT, diallyl sulfide, and α-tocopherol at 24 h of incubation in a Fe^2+^/H_2_O_2_ system may be due to the stronger protein oxidation, which is believed to result in protein fragmentation and degradation of the structural protein [5,46]. The result of SDS-PAGE was consistent with the changes in the carbonyl content (Figure 2).

### 3.7. Instability Index

The LUMiSizer system is utilized to evaluate the physical stability of MPs by monitoring the transmission changes due to flocculation. The more changes in the transmission during centrifugation, the lower stability of the MP droplets. The position in the transmission profiles at about 105 mm corresponded to the filling height of the MP solutions, and the position of the cuvette bottom was at 130 mm. All of the first profiles lay at the bottom as the red ones, and the last profiles lay at the top as the green ones [47]. It was observed from analyzing the transmission profiles (Figure 6A) that the sedimental steps were much faster in the oxidized control, which indicated the sedimentation velocity was higher after LUMiSizer centrifugation. The presence of antioxidants could slow the sedimentation rate and improve the physical stability of the MP solutions.

The instability index was employed to evaluate the physical stability of MPs. Low values of the instability index represent greater stability [32]. Figure 6B shows the oxidation time and antioxidant types on the instability index of MPs. With the oxidation time elongating from 12 h to 24 h, for all the treatments, the instability index increased significantly (*p* < 0.05). This suggested that oxidation reduced the physical stability of MPs in yak meat. Jayaraman et al. [48] reported that protein oxidation destroyed the integrity of the protein structure and accelerated protein unfolding, subsequently resulting in an unstable interface being formed. The cross-linking and aggregation of protein fragments due to protein oxidation, and eventually, the formation of large and insoluble aggregates, also likely led to protein instability [17].

The addition of antioxidants improved the physical stability of MPs, and rutin- and ferulic-acid-stabilized MPs had the best physical stability among the investigated antioxidants, followed by α-tocopherol. This was attributed to the fact that antioxidants stabilizing the oxidative environment by trapping free radicals contributed to decreased protein aggregates. In light of the report of Walstra [49], high electrostatic repulsion prevented the coalescence of dispersions by strengthening the repulsive force between the droplets; therefore, part of the improved stability may be attributed to the relatively strong electrostatic repulsion between the droplets after antioxidant treatment. However, it was noteworthy that BHT showed a relatively lower stability than natural antioxidants (*p* < 0.05), except for diallyl sulfide, which exhibited similar instability index as BHT (*p* > 0.05). The lower physical stability of diallyl sulfide and BHT incorporated treatments was closely related to their poor protection against droplet aggregation.

## 4. Conclusions

The results of this study suggested that four natural antioxidants exhibited high potential in protecting lipids and proteins of yak meat against oxidation, although different natural antioxidants showed significantly diverse behaviors in this regard. Among them, both ferulic acid and rutin were proven to be more effective in retarding lipid and protein oxidation than diallyl sulfide and α-tocopherol. Rutin, followed by ferulic acid, α-tocopherol, and diallyl sulfide, orderly demonstrated strong capability to physically stabilize the myofibrillar proteins of yak meat. Ferulic acid and rutin may be promising natural antioxidants for preventing oxidation of lipids and proteins in yak meat, and the possible protective mechanism of diallyl sulfide requires more research to draw an affirmable conclusion.

## Figures and Tables

**Figure 1 foods-11-03062-f001:**
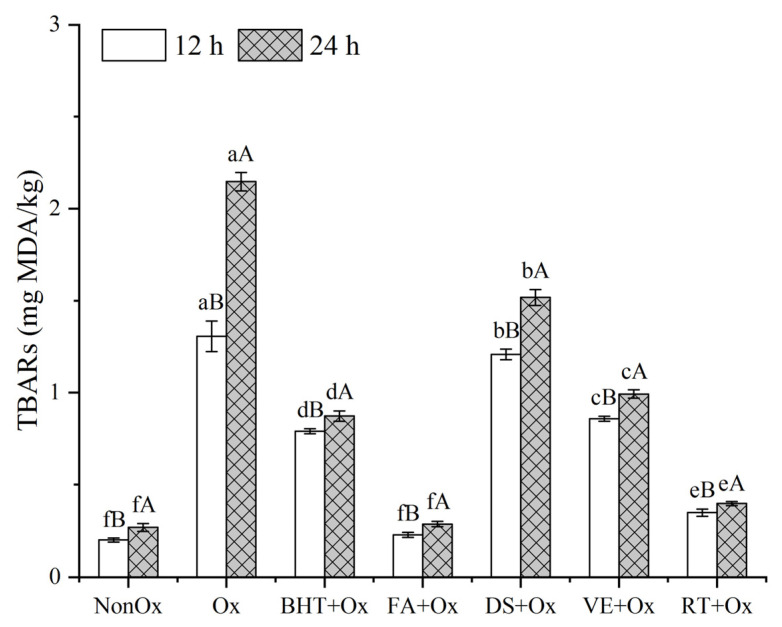
TBARs value of minced yak meat supplemented with different antioxidants. Note: NonOx—non-oxidized; Ox: oxidized without antioxidant; BHT + Ox, FA + Ox, DS + Ox, VE + Ox, and RT + Ox—oxidized in the presence of BHT, ferulic acid, diallyl sulfide, α-tocopherol, and rutin, respectively. ^a–f^ Means with different lowercase letters are significantly different among antioxidant treatments for the same oxidation time (*p* < 0.05); ^A,B^ Means with different uppercase letters are significantly different between 12 and 24 h for the same antioxidant treatment (*p* < 0.05).

**Figure 2 foods-11-03062-f002:**
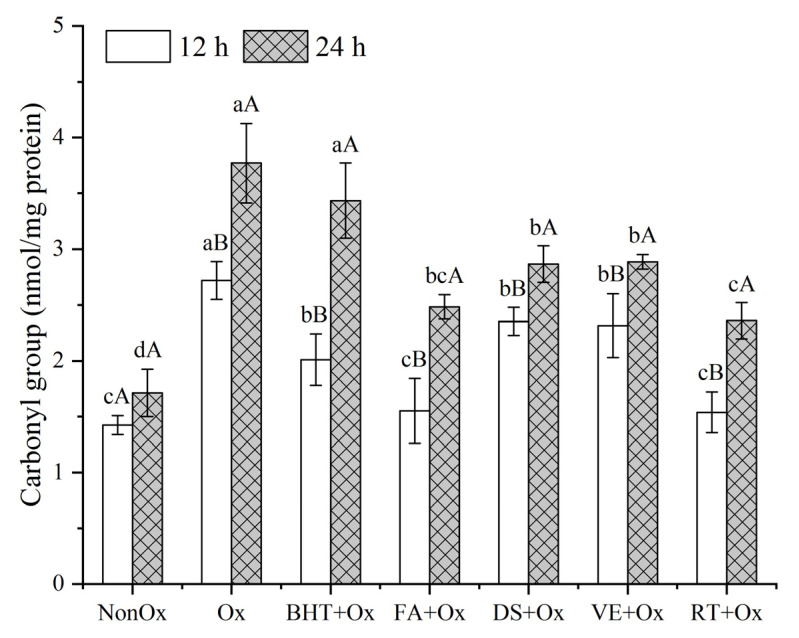
Protein carbonyl content of MPs in minced yak meat supplemented with different antioxidants. Note: NonOx—non-oxidized; Ox—oxidized without antioxidant; BHT + Ox, FA + Ox, DS + Ox, VE + Ox, and RT + Ox—oxidized in the presence of BHT, ferulic acid, diallyl sulfide, α-tocopherol, and rutin, respectively. ^a–d^ Means with different lowercase letters are significantly different among antioxidant treatments for the same oxidation time (*p* < 0.05); ^A,B^ Means with different uppercase letters are significantly different between 12 and 24 h for the same antioxidant treatment (*p* < 0.05).

**Figure 3 foods-11-03062-f003:**
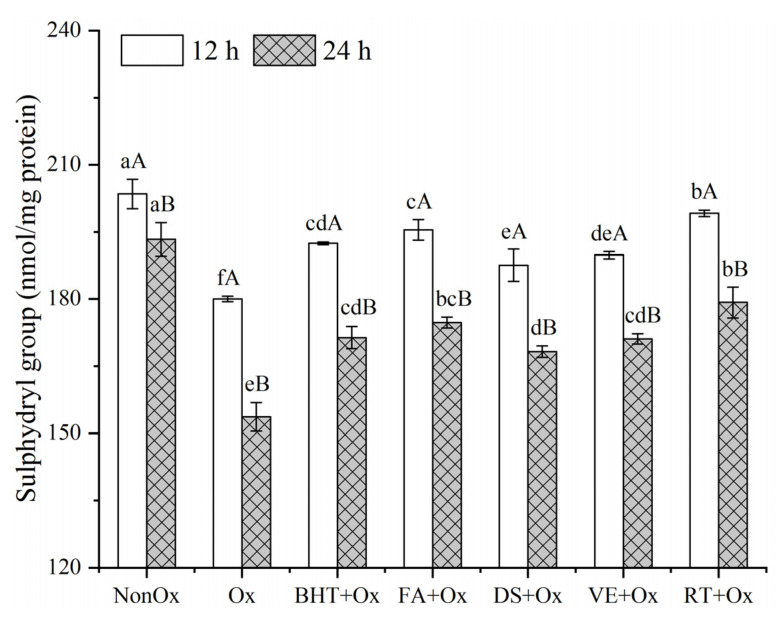
Protein sulfhydryl group of MPs in minced yak meat supplemented with different antioxidants. Note: NonOx—non-oxidized; Ox—oxidized without antioxidant; BHT + Ox, FA + Ox, DS + Ox, VE + Ox, and RT + Ox—oxidized in the presence of BHT, ferulic acid, diallyl sulfide, α-tocopherol, and rutin, respectively. ^a–f^ Means with different lowercase letters are significantly different among antioxidant treatments for the same oxidation time (*p* < 0.05); ^A,B^ Means with different uppercase letters are significantly different between 12 and 24 h for the same antioxidant treatment (*p* < 0.05).

**Figure 4 foods-11-03062-f004:**
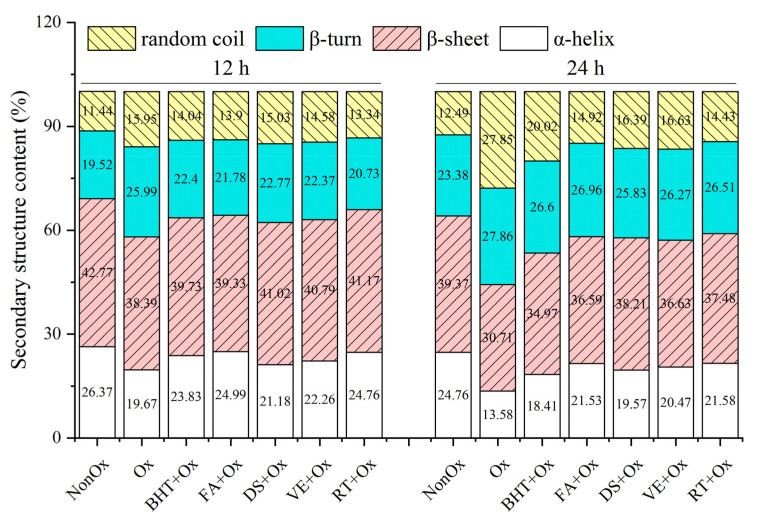
Secondary structural composition of MPs in minced yak meat supplemented with different antioxidants. Note: NonOx—non-oxidized; Ox—oxidized without antioxidant; BHT + Ox, FA + Ox, DS + Ox, VE + Ox, and RT + Ox—oxidized in the presence of BHT, ferulic acid, diallyl sulfide, α-tocopherol, and rutin, respectively.

**Figure 5 foods-11-03062-f005:**
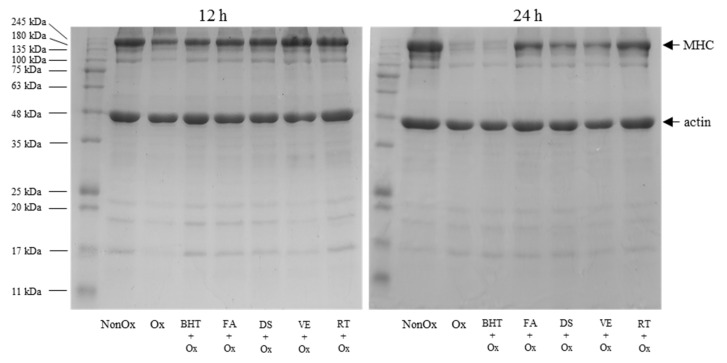
SDS-PAGE pattern of MPs in minced yak meat supplemented with different antioxidants. Note: NonOx—non-oxidized; Ox—oxidized without antioxidant; BHT + Ox, FA + Ox, DS + Ox, VE + Ox, and RT + Ox—oxidized in the presence of BHT, ferulic acid, diallyl sulfide, α-tocopherol, and rutin, respectively.

**Figure 6 foods-11-03062-f006:**
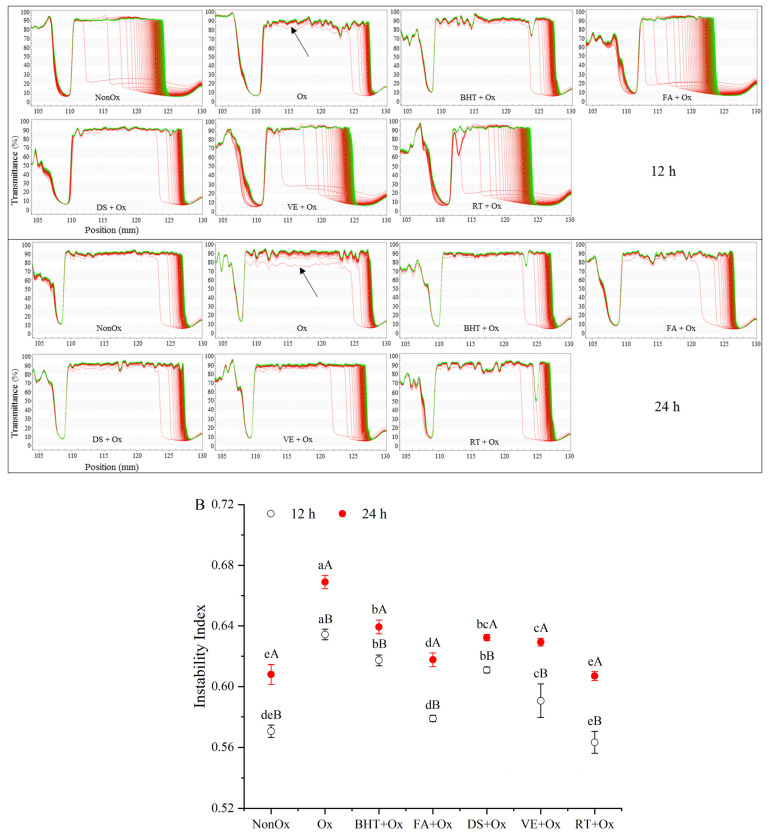
Physical stability profiles (original transmission) (**A**) and instability index (**B**) of MPs analyzed by LUMISizer at 12 h and 24 h in minced yak meat supplemented with different antioxidants. Note: NonOx—non-oxidized; Ox—oxidized without antioxidant; BHT + Ox, FA + Ox, DS + Ox, VE + Ox, and RT + Ox—oxidized in the presence of BHT, ferulic acid, diallyl sulfide, α-tocopherol, and rutin, respectively. ^a–e^ Means with different lowercase letters are significantly different among antioxidant treatments for the same oxidation time (*p* < 0.05); ^A,B^ Means with different uppercase letters are significantly different between 12 and 24 h for the same antioxidant treatment (*p* < 0.05).

**Table 1 foods-11-03062-t001:** POV and CD values of minced yak meat supplemented with different antioxidants.

Treatment	POV (meq/kg)		CD (mmol/kg)	
	12 h	24 h	12 h	24 h
NonOx	0.11 ± 0.03 ^bB^	0.18 ± 0.03 ^dA^	0.47 ± 0.06 ^cA^	0.51 ± 0.07 ^dA^
Ox	0.24 ± 0.03 ^aB^	0.34 ± 0.03 ^aA^	0.78 ± 0.05 ^aA^	0.88 ± 0.08 ^aA^
BHT + Ox	0.09 ± 0.03 ^bcB^	0.23 ± 0.01 ^bcA^	0.66 ± 0.06 ^bB^	0.82 ± 0.07 ^abA^
FA + Ox	0.05 ± 0.02 ^cB^	0.14 ± 0.03 ^dA^	0.46 ± 0.06 ^cB^	0.64 ± 0.05 ^cA^
DS + Ox	0.10 ± 0.03 ^bB^	0.25 ± 0.01 ^bA^	0.66 ± 0.04 ^bB^	0.74 ± 0.01 ^bA^
VE + Ox	0.06 ± 0.02 ^bcB^	0.19 ± 0.03 ^cdA^	0.64 ± 0.07 ^bA^	0.75 ± 0.03 ^bA^
RT + Ox	0.07 ± 0.02 ^bcB^	0.17 ± 0.03 ^dA^	0.46 ± 0.04 ^cB^	0.56 ± 0.03 ^cdA^

Note: Data were presented as mean ± SD. NonOx—non-oxidized; Ox—oxidized without antioxidant; BHT + Ox, FA + Ox, DS + Ox, VE + Ox, and RT + Ox—oxidized in the presence of BHT, ferulic acid, diallyl sulfide, α-tocopherol, and rutin, respectively. ^a–d^ Means with different lowercase letters are significantly different among antioxidant treatments for the same oxidation time (*p* < 0.05); ^A–B^ Means with different uppercase letters are significantly different between 12 and 24 h for the same antioxidant treatment (*p* < 0.05).

## Data Availability

The data presented in this study are available on request from the corresponding author.

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
