# Peer review of "Protective Effects of Four Natural Antioxidants on Hydroxyl-Radical-Induced Lipid and Protein Oxidation in Yak Meat"

_foods, 2022, doi:10.3390/foods11193062_

Round 1

Reviewer 1 Report

The manuscript is about the comparison between the antioxidant effect of natural antioxidants against BHT in Yak meat.

This subject is fully covered by the literature in many types of meat and meat products using a variety of natural antioxidants, however this study shows relevancy thanks to different choices made by the authors regarding the experimental plan. Meanwhile, some corrections and clarifications must be answered to go further with this work.

Abstract:

Please specify the exact meaning of the abbreviations in the abstract (POV, CD, etc.)

Introduction:

Lines from 24 to 26: Please add a reference to the statement in the sentence.

Materials and Methods

Line 83: Please specify what are the other chemicals and reagents here. The authors can add a reference to the following paragraphs for instance.

Line 90 and 91: Why did the authors chose to use a constructed oxidation system? It would have been more suitable to put the product in chilled storage and study the naturally occurring oxidation during that period. The formulations would be under the same conditions to mimic normal natural conditions. Please explain your choice.

Lines from 86 until 99: How many times the whole experiment was repeated?

Lines 192 until 196: Explain more in detail why this statistical methodology was used? And specify the type of the statistical analysis: descriptive?

Reviewer 2 Report

Abstract and introduction- explain abbreviation in first use

L 91 - explain in detail oxidation system

L 93 - muscle or sample?

L 81 - how much meat did you use for sampling, and the avearage weight of samples was?

L 210 - please explain in short findings of Maqsood et al.

L 348 - please explain in short result of mentioned study

Please explain (I suggest in the introduction) where this potential of natural antioxidants could be used - I know- meat industry - but please give some examples.
